# Advancements in Spinal Cord Injury Repair: Insights from Dental-Derived Stem Cells

**DOI:** 10.3390/biomedicines12030683

**Published:** 2024-03-19

**Authors:** Xueying Wen, Wenkai Jiang, Xiaolin Li, Qian Liu, Yuanyuan Kang, Bing Song

**Affiliations:** 1School and Hospital of Stomatology, China Medical University, Liaoning Provincial Key Laboratory of Oral Diseases, Shenyang 110002, China; wenx14@cardiff.ac.uk (X.W.); lixiaolin@cmu.edu.cn (X.L.); 20082068@cmu.edu.cn (Y.K.); 2State Key Laboratory of Oral & Maxillofacial Reconstruction and Regeneration, National Clinical Research Center for Oral Diseases, Shaanxi Key Laboratory of Stomatology, Department of Operative Dentistry & Endodontics, School of Stomatology, Fourth Military Medical University, Xi’an 710032, China; jiangw6@cardiff.ac.uk; 3Shenzhen Institutes of Advanced Technology, Chinese Academy of Sciences, Shenzhen 518055, China; liuqian@siat.ac.cn; 4School of Dentistry, Cardiff University, Heath Park, Cardiff CF14 4XY, UK

**Keywords:** spinal cord injury, stem cell therapy, neural repair, dental-derived stem cells, dental pulp stem cells, stem cells from human exfoliated deciduous teeth, stem cells from the apical papilla, dental follicle stem cells, delivery system, tissue regeneration

## Abstract

Spinal cord injury (SCI), a prevalent and disabling neurological condition, prompts a growing interest in stem cell therapy as a promising avenue for treatment. Dental-derived stem cells, including dental pulp stem cells (DPSCs), stem cells from human exfoliated deciduous teeth (SHED), stem cells from the apical papilla (SCAP), dental follicle stem cells (DFSCs), are of interest due to their accessibility, minimally invasive extraction, and robust differentiating capabilities. Research indicates their potential to differentiate into neural cells and promote SCI repair in animal models at both tissue and functional levels. This review explores the potential applications of dental-derived stem cells in SCI neural repair, covering stem cell transplantation, conditioned culture medium injection, bioengineered delivery systems, exosomes, extracellular vesicle treatments, and combined therapies. Assessing the clinical effectiveness of dental-derived stem cells in the treatment of SCI, further research is necessary. This includes investigating potential biological mechanisms and conducting Large-animal studies and clinical trials. It is also important to undertake more comprehensive comparisons, optimize the selection of dental-derived stem cell types, and implement a functionalized delivery system. These efforts will enhance the therapeutic potential of dental-derived stem cells for repairing SCI.

## 1. Introduction

Spinal Cord Injury (SCI) is a highly intricate and catastrophic neurological disorder that often results in temporary or permanent changes in its function, leading to a range of motor, sensory, and vegetative dysfunctions, making it one of the most complex and devastating nervous system disorders [1]. The causes of SCI encompass a variety of factors, such as traffic accidents, violent injuries, falls, degenerative diseases of the spine, tumors, infections, ischemia-reperfusion injuries, and blood vessel-related injuries [2]. SCI can be categorized into traumatic and non-traumatic SCI, and from a pathophysiological perspective, acute SCI can further be categorized into primary injury and secondary injury. Additionally, the severity of SCI can be distinguished as complete injury and incomplete injury [3].

Despite advancements in modern medicine that have improved the survival rates of SCI patients, progress in alleviating functional impairments related to neurogenic shock, respiratory difficulties, changes in ion and neurotransmitter levels, and inflammation remains limited, resulting in a poor quality of life for SCI patients [4]. Moreover, SCI imposes a substantial economic burden on patients, their families, and communities. Over the past 30 years, the global incidence of SCI has increased from 236 cases per million people to 1298 cases [5]. It is estimated that the annual global incidence of SCI ranges from 250,000 to 500,000 individuals [6]. According to the 2016 SCI Data Sheet published by the National Spinal Cord Injury Statistical Centre in the United States, healthcare costs and living expenses for SCI patients in the first year can amount to as high as $1,065,980, with average annual indirect costs, such as lost wages, additional benefits, and productivity losses, at $72,047 [7].

Alongside the surgical intervention in treating SCI, the current treatment methods for SCI also involve using anti-inflammatory medications such as ketorolac, minocycline, riluzole, magnesium, decompression surgery (decompression and instrumentation) to stabilize the spinal column and proper supportive management to prevent secondary injury [8]. Based on the current literature, no guidelines strongly endorse surgical intervention or pharmacological treatment as the primary method for treating SCI. Nevertheless, the treatment outcomes and prognosis for SCI patients are far from ideal [9]. Consequently, SCI remains a global challenge in clinical settings, presenting a significant hurdle for neuroscientists and neurosurgeons alike [10].

In the last decade, stem cell (SC) therapy has emerged as a novel and promising treatment for SCI. Stem cells derived from dental sources, such as dental pulp stem cells (DPSCs), stem cells from human exfoliated deciduous teeth (SHED), stem cells from the apical papilla (SCAPs), dental follicle stem cells (DFSCs), have garnered significant attention due to their ease of procurement and robust neurogenic differentiation potential. There is a promising outlook for their application in SCI treatment. Within the scope of this review, we comprehensively examined how dental stem cells promote SCI repair through various methods, including stem cell transplantation, conditioned culture medium injection, bioengineered delivery systems, exosomes, and extracellular vesicle treatments, and combined therapies, and discussed the limitations of existing research, offering new insights for future research directions. Through a meticulous examination of methodologies and a critical discussion of existing research, we not only highlight the innovative potential of dental-derived stem cells in SCI treatment but also provide new perspectives for future research directions and treatment methods.

## 2. Phases and Pathophysiology of SCI

The degree of loss of neurological function is measured using the American Spinal Injury Association (ASIA) impairment scale: A (complete sensory and motor loss below the lesion, including sacral sensation loss), B (sensory function below the lesion with no motor function), C (partial motor function preservation, with most muscles grading below 3), D (more than half of muscles below the lesion grading at three or higher), and E (normal motor and sensory testing) [11]. The AISA impairment scale classifies spinal cord injuries into complete and incomplete types. Complete injuries are Grade A, while incomplete injuries are graded B through D [12]. According to studies reporting complete SCI (grade A), there is little chance of resuming standing or walking with exercise training alone [13]. In chronic patients with incomplete SCI who are grade C or D and more than 2 years post-injury, the results of reconstructing gait with manually assisted exercise training with weight support are also unsatisfactory [14]. At the moment, incomplete tetraplegia is the most common SCI type (45%), followed by incomplete paraplegia (21.3%), complete paraplegia (20%), and complete tetraplegia (13.3%) [15]. Less than 1% of patients achieve complete recovery after discharge [15]. Spontaneous recovery after SCI is exceedingly rare, possibly due to several inhibitory regulatory factors, such as extracellular matrix proteins, which reduce the spinal cord’s potential for endogenous repair by lowering its regenerative and plasticity capacities [2].

The pathological mechanisms of SCI involve two primary processes: primary injury and secondary injury (Figure 1). Primary injury occurs when the spinal cord is exposed to external forces such as contusion, ripping, compression, or transection or when it suffers an ischemic infarction due to vascular damage [16]. The features of the primary pathology of SCI are bone fragments and spinal tissue tearing. In primary injury, the pathophysiology includes neural parenchyma, axonal network glial membrane disruption, and hemorrhage (Figure 1) [2,17]. The severity of the injury is determined by the extent of initial destruction and the duration of spinal cord compression. A cascade of events associated with secondary injury is triggered by biochemical, mechanical, and physiological changes within neural tissues [18].

Following primary injury, a retrograde process initiates within minutes to hours, and its severity is directly proportional to the extent of the initial injury. Secondary injury is classified into three phases: acute, sub-acute, and chronic (Figure 1). During the acute phase, which occurs within 0–48 h, the spinal cord experiences excitotoxicity, vascular damage, ionic imbalance, increased calcium influx, edema, free radical production, inflammation, lipid peroxidation, and necrosis [2,19]. The acute phase persistence leads to the sub-acute phase, characterized by glial scarring, neuronal apoptosis, axonal demyelination, Wallerian degeneration, and axonal remodeling in the first two weeks [2,20]. During the chronic secondary injury phase of SCI, extending from days to years, cystic cavity forms, axonal dieback occurs, and the glial scar matures [21,22]. These three phases in secondary injury cause damage to underlying nucleic acid, proteins, and phospholipids, resulting in neurological dysfunction [5].

Besides, it is imperative to consider the role of imaging in assessing and prognosticating these injuries. Recent studies have highlighted the utility of magnetic resonance imaging (MRI) in providing valuable insights into the pathophysiology and prognosis of SCI [23]. For instance, MRI intramedullary signal characteristics in the early stages after SCI have been shown to correlate with the severity of injury and predict functional recovery [24].

Models of SCI are classified by the injury mechanism as contusion, compression, distraction, dislocation, transaction, or chemical. Revealing their characteristics helps us understand the unique pathophysiology of various SCI models (Table 1).

Interventions for enhancing SCI recovery aim to minimize the spread of secondary injury (through neuroprotection or inflammation modulation) and to replace lost nerve cells and disrupted neural circuits (through neuroplasticity and regeneration) [34]. These interventions seek to maximize the patient’s rehabilitation potential and alleviate the functional impairments caused by SCI.

## 3. The Neurodegenerative Potential of Dental Stem Cells

Stem cells, with their pluripotent nature, proliferate and self-renew under specific conditions [35]. Mesenchymal stromal cells (MSCs), a diverse group of postnatal stem cells, stand out for their unique characteristics, including self-renewal, multipotent differentiation capabilities, and immune system modulation [36]. Extensive studies in the last two decades have emphasized MSCs’ pivotal role in tissue homeostasis, with applications ranging from treating autoimmune diseases to regenerating damaged tissues like SCI [37,38,39,40].

MSCs have been successfully isolated from various adult and neonatal tissues, including bone marrow, skin, dental, adipose tissue, umbilical cord, Wharton’s jelly, and placenta [38,41,42,43,44]. Dental pulp tissue-derived MSCs, notably, offer distinct advantages over other MSC sources, such as bone marrow, adipose tissue, peripheral blood, and umbilical cord blood, due to their ease of accessibility, good proliferation potential, neurogenic differentiation and neurotrophic capabilities, and negligible ethical issues and minimal invasiveness [45]. They are well-suited for tissue engineering and gene therapy due to their high proliferative potential, regenerative capacity, ability to differentiate into multiple cell types and lower inherent immunogenicity [45].

Dental stem cells can be isolated from different tooth regions, including DPSCs from the pulp of third molars, DFSCs from the dental follicle membrane that surrounds developing teeth, SHED from children’s shed deciduous teeth, and SCAP from immature teeth (Figure 2) [46,47,48,49,50]. They all possess a certain degree of neurogenic differentiation capability.

### 3.1. DPSCs

DPSCs, first derived from the dental pulp in 2000 by Gronthos [51], share characteristics with mesenchymal stem cells, including plasticity, adhesiveness, and a fibroblast-like morphology [45]. Notably, DPSCs express neurotrophic and immunomodulatory factors, promoting blood vessel formation and nerve regeneration [52].

Various studies highlight DPSCs’ potential for neurodegeneration. Recent research demonstrates that combining vascular endothelial growth factor A (VEGFA)-overexpressing rat dental pulp stem cells (rDPSCs) with a laminin-coated and yarn-encapsulated poly (l-lactide-co-glycolide) (PLGA) nerve guidance conduit (LC-YE-PLGA NGC) enhances myelin sheath quantity, thickness, and axonal diameter, facilitating the repair of facial nerve injuries [53]. Similar experiments show that chitosan tubes inoculated with stem cell factor and DPSCs boost neo-vascularization, providing an effective approach to repairing facial nerve defects [54]. In a rodent model with sciatic nerve deficits, transplantation of neuro-lineage cells (NLC) differentiated from DPSCs revealed substantial improvements in axonal growth, remyelination, electrophysiological activity, and muscle atrophy after 12 weeks [55]. Ben Mead and colleagues’ study demonstrates that intravitreal transplants of DPSCs significantly enhance neurotrophin-mediated retinal ganglion cell (RGC) survival and axon regeneration following optic nerve injuries [56].

### 3.2. SHED

SHED, a unique subtype of pluripotent stem cells, was initially isolated and characterized from the pulp tissue of shed human deciduous teeth by Miura and colleagues [48]. Acknowledged for its highly proliferative capacity and ability to differentiate into various cell types [57]. SHED demonstrates persistence within the mouse brain and expression of neural markers upon in vivo transplantation. Derived from easily accessible tissue sources, SHED provides an abundant cell supply for potential clinical applications [48].

Both SHED and their conditioned media (SHED-CM) effectively address neurodegenerative diseases through mechanisms such as cell replacement, paracrine effects, angiogenesis, synaptogenesis, immunomodulation, and inhibition of apoptosis [58]. SHED’s neurogenic differentiation potential and release of bioactive molecules offer promise for neuronal recovery in nerve injury cases, holding potential for disorders like SCI, Alzheimer’s disease (AD), and focal cerebral ischemia (FCI) [59]. Research by Sugimura and colleagues highlights SHED-CM’s potential in promoting axonal regeneration and functional recovery in a rat model with sciatic nerve deficits. Their findings suggest that SHED-CM facilitates axonal growth, peripheral nerve tissue vascularization, neuronal survival, and Schwann cell migration and proliferation [60].

### 3.3. SCAP

In 2006, Sonoyama and colleagues cultured cells from apical papilla tissue collected from the roots of young human third molars, identifying them as MSCs and naming them SCAP [61]. SCAP, derived from the apical papilla, is more accessible, separable, and expansible compared to other dental tissues, such as DPSCs [62]. SCAP exhibits superior proliferation rates and expresses classical stem cell markers, outperforming DPSCs in BrdU uptake rate, cell doubling, tissue regeneration capacity, and the number of STRO-1-positive cells [63]. Co-culturing SCAP with trigeminal sensory neurons enhances sustained inward current density induced by ATP, suggesting a positive impact on sensory nerve activity and cold-sensitive ion channels [64]. SCAP also demonstrates low immunogenicity and possesses immunomodulatory characteristics [65]. Studies indicate their role in expediting SCI healing by reducing TNF-α levels and promoting oligodendrocyte progenitor cell differentiation [66]. Consequently, the apical papilla and its resident SCAP offer a unique opportunity for their potential clinical application in neural repair.

### 3.4. DFSCs

Dental follicle cells (DFCs) were first reported by Wise et al. in 1992, and later, in 2005, a population of cells with colony-forming and plastic-adherent properties was successfully derived from dental follicles, termed dental follicle progenitor/stem cells (DFPCs/DFSCs) [67,68]. As part of the Dental Stem Cell family, DFSCs are obtained during the early stages of development, providing an advantageous cell source for clinical applications due to the larger tissue volume of dental follicles.

DFSCs exhibit a higher proliferation rate, enhanced colony-forming capability, and potent anti-inflammatory characteristics compared to other dental MSCs, making them clinically relevant for treating oral and neurological disorders [69]. Originating from the cranial neural crest, DFSCs express neurogenic membrane markers like NESTIN and TUBULIN IIIβ, and DFCs retain multipotential differentiation, displaying neurogenesis-related behaviors. Obtained easily from third molar extraction or alveolar fossa curettage, human DFCs (hDFCs) are more inclined to express the neurogenic marker MAP2 compared to SHEDs, suggesting their efficiency in neural regeneration [70,71].

In summary, dental-derived stem cells, irrespective of their origin, exhibit rapid proliferation and the ability to differentiate into typical mesenchymal cell lineages, including osteo/dentinogenic, adipogenic, and neurogenic pathways [72]. Derived from the neural crest, dental MSCs uniquely express neural markers, secrete neurotrophic factors, and combine MSC-like characteristics with immunomodulation and neural features [73]. With their neural crest lineage, neuronal markers, and expression of neurotrophic factors, along with the potential for neurogenic differentiation, these cells are actively researched for their applications in treating neuronal diseases and injuries [74,75]. Recent studies highlight their promising role in neural tissue engineering for nerve regeneration, particularly in the treatment of SCI.

## 4. Treatment Approaches

In the pursuit of effective SCI treatment, diverse approaches leveraging the potential of dental-derived stem cells have surfaced. Current research utilizes various strategies, including stem cell transplantation, conditioned culture medium injection, innovative stem cell delivery systems, exosome-based therapies, and combined interventions with physical methods. Collectively, these approaches constitute a multifaceted framework for SCI treatment (Figure 3).

### 4.1. Stem Cell Implantation

Comparative studies involving mesenchymal stem cells (DP-MSCs, AD-MSCs, and DP-MSCs) reveal that transplanting DP-SMCs into SCI sites effectively promotes neural regeneration and functional recovery [76]. Similarly, DPSCs and SHED transplantation into a completely transected mouse acute SCI model significantly enhanced motor function and promoted spinal cord axon regeneration [77]. DPSCs, through mechanisms like reducing cell apoptosis, promoting axon regeneration, and differentiating into mature oligodendrocytes, contribute to functional recovery after SCI [77].

In addressing the hypoxic environment in SCI, introducing the basic fibroblast growth factor (bFGF) gene into DPSCs via an adeno-associated virus (AAV) vector successfully ameliorates the hypoxic environment, aiding neuron survival and axon regeneration [78].

Additionally, experiments conducted by Fabrício Nicola and their research team have shown that injecting SHED cell suspensions into rodent SCI sites improves motor function, reduces tissue loss, protects neurons, mitigates inflammation, and decreases neuronal apoptosis [79]. SHED transplantation fosters neural precursor cell proliferation, reduces glial scar formation, and slows S100B protein decline in spinal glial cells [80].

Using induced pluripotent stem cells (iPSCs) derived from SHED (iSHED) enhances regenerative potential when transplanted into a rat model of acute SCI [81].

Furthermore, experiments led by Chao Yang and their research team found that various types of dental-derived stem cells, including DFSCs, SCAP, and DPSCs, show their potential to induce neural regeneration, reduce inflammatory responses, promote neural regeneration, and differentiate into mature neurons and oligodendrocytes [82]. DFSCs exhibited particularly significant effects.

In summary, direct transplantation of dental-derived stem cells shows promise in treating acute and chronic SCI, emphasizing their role in promoting neural regeneration and functional recovery. Further research and clinical trials are needed to determine optimal treatment methods and application areas.

### 4.2. Condition Medium Injection

Cell therapy for SCI poses risks of tumor formation and immune reactions. To overcome these challenges, researchers advocate for conditioned media (CM) from cell sources as an alternative. Dental pulp stem cells’ cell source-conditioned medium (DPSC-CM) and human shed deciduous teeth stem cells (SHED-CM) have demonstrated significant potential in restoring cerebellar granule neurons’ neurite growth activity, surpassing the effectiveness of conditioned media from fibroblasts or bone marrow mesenchymal stem cells [62].

SHED-CM, in particular, induces an M2-dominant neural repair microenvironment and enhances functional recovery after SCI through factors like MCP-1 and ED-Siglec-9 [83].

To address concerns about the rapid spread of CM through body fluids, a combination method involving biological materials and drug delivery systems has been proposed. A study by Reza Asadi-Golshan’s team utilized a collagen hydrogel as a slow-release vehicle for SHED-CM, demonstrating significant advantages in various scores related to spinal cord injury recovery [84]. Another study by the same team found that injecting collagen hydrogel-loaded SHED-CM into the spinal cord after compression SCI in rats prevented tissue loss, including grey and white matter, and protected cells, including neurons and oligodendrocytes [85].

### 4.3. Bioengineered Delivery System Approaches

While stem cell therapy holds promise for tissue repair and regenerative medicine, challenges persist. Similar to organ transplantation, immunosuppressants are often required for enhanced cell viability and survival. Acute inflammation and immune reactions, along with the absence of extracellular matrix support, contribute to the early demise of transplanted cells [86]. Controlling the fate and integration of transplanted cells within the organism presents further challenges. Researchers are addressing these issues by exploring the immobilization of dental stem cells in biocompatible materials, such as hydrogels, chitosan, PLGA scaffolds, microcapsules, and microspheres, offering promising solutions for SCI treatments.

#### 4.3.1. Hydrogel

After SCI, the local microenvironment becomes detrimental due to the release of excitatory neurotransmitters and toxic substances [87]. To address this challenge, an innovative hydrogel known as TPA@Laponite hydrogel has been developed by Yigo Ying and colleagues. This shear-thinning hydrogel, encapsulating and protecting DPSCs, effectively scavenges harmful oxygen radicals, enhances vascular function, and inhibits lipid peroxidation. DPSCs introduced into this hydrogel adjust the excitatory to inhibitory synapse ratio, reducing muscle spasms and promoting SCI recovery [88].

Moreover, Heng Zhou addressed the loss of zinc ions (Zn^2+^) post-SCI by introducing ZIF-8 into DPSCs and injecting them into injured rat spinal cords. ZIF-8, a carrier for drug and gene delivery, releases Zn^2+^ in acidic environments. Combined with gelatin methacryloyl (GelMA) hydrogel, this approach promotes motor function recovery, facilitating neural cell repair, inhibiting apoptosis, and enhancing angiogenesis [89].

Pluronic F-127, a synthetic hydrogel, has been explored as an injectable carrier [90]. Lihua Luo’s team designed a thermosensitive heparin-poloxamer (HP) hydrogel containing bFGF and DPSCs. This hydrogel, delivered to the spinal cord injury site, sustains DPSC density and bFGF activity, promoting tissue regeneration and improving sensory and motor function recovery [91]. Another study suggests that heparin-based hydrogel containing bFGF and DPSCs (HeP-bFGF-DPSCs) effectively controls inflammation, stabilizes microtubules, regulates the tissue vascular system, and promotes neural regeneration [92].

Calcium alginate hydrogel, a biocompatible material, combined with DPSCs and fibroblast growth factor 21 (FGF21), demonstrated exceptional tissue affinity in Sipin Zhu’s study. This hydrogel protects neurons, inhibits apoptosis, promotes autophagy, and enhances recovery post-spinal cord transection surgery by facilitating axon and functional blood vessel regeneration [93].

Additionally, the research attempted to encapsulate the entire human dental pulp in a fibroin hydrogel. Dental pulp implantation in SCI reduced pro-inflammatory markers, inhibited microglia/macrophage activation, and increased immunoreactive 5-HT positive cells [94]. However, a study indicates that, compared to the implantation of the entire dental pulp, the effectiveness of using dental root apical papilla stem cells combined with hydrogel is inferior. Experimental evidence demonstrates that rats receiving dental pulp implantation achieved positive outcomes in terms of recovery, chronic pain, and spinal cord tissue structure [95].

#### 4.3.2. Chitosan

Research into chitosan-bound dental stem cells shows promising prospects. Chitosan, a highly biocompatible and hydrophilic biopolymer extracted from crustaceans’ exoskeletons, such as crabs, shrimps, and lobsters, is widely used in tissue engineering and various medical fields. Treatment involving DPSCs combined with a chitosan scaffold has shown enhanced cell viability and neural differentiation. In comparison to the control group without a chitosan scaffold, the DPSCs/chitosan scaffold group exhibited significantly elevated levels of BDNF, GDNF, b-NGF, and NT-3. Transplanting DPSCs with a chitosan scaffold into an SCI rat model resulted in a substantial recovery of hindlimb motor function, with the transplantation group experiencing significantly lower tissue loss, apoptotic cell count, and axon degeneration than other experimental groups. Research also identified the crucial role of the Wnt/β-catenin signaling pathway in neural differentiation when DPSCs were combined with a chitosan scaffold [96].

Additionally, studies have explored the combined use of a chitosan scaffold with bFGF and DPSCs [97]. Results indicate that the treatment group combining DPSCs/chitosan scaffold with bFGF had significantly higher levels of GFAP, S100b, and b-tubulin protein III compared to the control group without bFGF and the group using only DPSC/chitosan scaffold. This suggests that the combined application of a chitosan scaffold with bFGF promotes the neural differentiation of DPSCs. Therefore, the transplantation of DPSCs/chitosan scaffold combined with bFGF presents a potential as a safe and effective treatment for SCI.

#### 4.3.3. PLGA

PLGA (poly-lactic-co-glycolic acid) scaffolds are extensively studied for supporting dental stem cell transplantation in SCI treatment. The limited regeneration of damaged spinal cords is attributed to inadequate vascular supply and neural nutritional support. To address this, researchers led by Shaowei Guo designed a highly vascularized scaffold using biocompatible and biodegradable poly (L-lactic acid) (PLLA)/PLGA scaffolds. Their results demonstrate that this scaffold, containing DPSCs, has the potential to enhance SCI repair through paracrine-mediated angiogenesis and neural regeneration. Implanting these scaffolds into the injured spinal cord of rats with complete spinal cord transection models promotes vascular reconstruction, aiding in axon regeneration, myelin deposition, and sensory recovery. Analysis of the reconstructed spinal cord tissue using 3D micro-computed tomography (micro-CT) imaging and morphometric measurements revealed a substantial presence of regenerating blood vessels, particularly in the sensory tract area, correlating with behavioral recovery after pre-vascularization treatment [98].

For cases involving relatively large spinal cord defects (SCD), which may cause interruptions or blockages in neural pathways requiring reconnection and reconstruction, a study utilized oriented electrospun PCL/PLGA materials (AEM). It was demonstrated that human dental follicle cells (hDFCs) could effectively grow along these oriented fibers. Although observations regarding functional recovery did not significantly differ among the groups, the implanted AEM-hDFCs composite material promoted the migration and growth of neural cells [99].

#### 4.3.4. Microcapsules and Microspheres

To precisely control cell migration, differentiation, and tissue integration, researchers developed microcapsules containing dental stem cells. One study indicated that biocompatible microcapsules could control the fate of dental stem cells in situ to promote the replacement of damaged or lost tissues in SCI. Lorena Hidalgo San Jos and colleagues utilized a microfluidic device to encapsulate DPSCs in alginate-collagen microcapsules, demonstrating cell survival for up to 21 days. The transplanted microcapsules effectively retained DPSCs in an organotypic SCI model, with the cells expressing neural markers after the in situ culture [100].

In another study, researchers encapsulated recombinant BDNF nano-precipitates in PLGA-P188-PLGA microspheres (BDNF-PAM) and implanted SCAP-derived stem cells into PAM. The SCAP BDNF-PAM treatment significantly increased cell retention in the spinal cord, improved motor coordination, reduced inflammation, and promoted axon growth, marking a novel injectable cell delivery system’s benefits in the SCI treatment [101].

In summary, combining hydrogels, chitosan scaffolds, PLGA scaffolds, and carriers like microcapsules and microspheres with dental stem cells holds promising prospects for SCI therapy. These methods offer opportunities to enhance cell survival, promote neural regeneration, inhibit inflammation, and facilitate tissue repair, ultimately restoring motor and sensory functions. Their biocompatibility and biodegradability make these carriers ideal for creating a conducive environment in damaged spinal cords, fostering neuron regeneration, and inducing cell differentiation to improve motor function.

### 4.4. Exosomes and Extracellular Vesicles

Extracellular vesicles derived from stem cells have gained prominence in spinal cord injury treatment. Produced by living cells, these vesicles contain proteins crucial for immune modulation, neuroprotection, and cellular behavior orchestration [102]. Unlike traditional stem cell therapy, stem cell-derived extracellular vesicles offer advantages such as smaller size, reduced tumorigenicity, enhanced membrane transfer efficiency, and the ability to traverse the blood-spinal cord barrier [103].

Chao Liu’s work highlights DPSCs-derived exosomes’ potential in attenuating M1 macrophage polarization by inhibiting the ROS-MAPK-NFκB P65 signaling pathway. This leads to reduced inflammation and neural damage in SCI, improving the recovery process. DPSCs-derived exosomes emerge as a promising therapeutic modality for spinal cord injuries, offering a potential strategy for mitigating secondary damage by disrupting the ROS and M1 macrophage polarization feedback loop [104].

### 4.5. Combined Therapies

In the pursuit of effective SCI treatment, dental-derived stem cells have recently gained significant attention as a promising therapeutic tool combined with physical Interventions. This multimodal approach integrates dental-derived stem cells with treatments like underwater treadmill therapy, treadmill training therapy, and electroacupuncture, offering increased hope for SCI patients.

Matheus Levi Tajra Feitosa and colleagues selected three canine breeds with chronic SCI-induced paralysis, meeting specific inclusion criteria like paraplegia, lack of conscious proprioception, exaggerated reflexes, and absence of deep pain sensation. Confirmed through MRI diagnosis of thoracolumbar intervertebral disc disease, the researchers injected immature dental pulp stem cells at three points into the SCI site. Over two months post-surgery, the animals underwent weekly veterinary physical therapy, including hydrotherapy with underwater treadmill sessions. Clinical assessments using the Olby scoring system revealed improvements, indicating the potential positive impact of combining stem cell therapy and physical treatments on chronic SCI recovery. However, precise mechanisms require further investigation [105].

While some studies found that combination therapy did not yield significant effects, César Prado and colleagues evaluated the safety, feasibility, and therapeutic effects of canine exfoliated deciduous tooth stem cell transplantation combined with electroacupuncture for chronic SCI in dogs. Results showed only mild improvements in some animals, with no significant therapeutic effects observed with stem cell treatment, electroacupuncture, or their combination. This could be attributed to the limited number of animals and significant variability in spinal cord injury severity [106]. Additionally, combining SHED transplantation with treadmill training did not yield significant improvements in motor function recovery in traumatic SCI rats compared to using SHED alone, highlighting the need for further research to determine the optimal timing and intensity of exercise training [107].

These studies shed light on the application of dental-derived stem cells in combination with other treatment modalities for SCI therapy, each with its advantages and disadvantages (Table 2). While providing valuable insights, further comprehensive research is essential to fully understand the efficacy and potential of these treatments.

## 5. Discussion

### 5.1. Limitations of Treatment Methods

Current research in dental-derived stem cells for SCI treatment shows promising advances but faces notable constraints, including model choice, the role of stem cells, and delivery system safety.

A Primary limitation is the predominant focus on murine models, with a lack of large-animal studies and clinical trials to validate therapy feasibility and efficacy. For instance, DP-MSC application in porcine SCI models did not yield motor function recovery comparable to murine models [110]. Some studies use small sample sizes, potentially compromising statistical significance. The complexity of SCI models makes them challenging to control, leading to variations and inconsistencies [99].

Different dental stem cell sources, such as DPSCs, SHED, and SCAP, lack distinctly delineated advantages and applicability in SCI treatment. Comparative analysis showed that DFSCs were more effective in SCI repair than DPSCs and SCAP [82]. Disparities between sensory and motor aspects and incongruities between clinical assessments and imaging results pose challenges [98,105]. The nonlinearity of the BBB score complicates discerning improvement degrees, especially in patients with substantial motor function [107]. Despite increased neural markers in some in vitro experiments, more in vivo experiments are needed to confirm neural cell replacement [101]. Further research is needed to elucidate how dental stem cells comprehensively promote neuroregeneration, optimal strategies for combination with other modalities, and the biological safety of co-applying with biomaterials [101] Studies on SCAP and hydrogels show inconclusive results, contrasting with SHED treatment [95].

CaCl_2_ application in isolation and concentrations in hydrogels need careful assessment [93]. Studies focused on short-term effects necessitate more comprehensive research, including extended follow-ups, to address potential adverse consequences like teratoma and tumorigenesis [78,81,92].

In addition, 11.8% of SCIs and 19.2% of non-traumatic SCIs have been reported to be caused by spinal tumors [111]. As an exceptional case, spinal tumors that cause SCI are particularly challenging to treat, and early diagnosis, multidisciplinary care, and appropriate rehabilitation are essential to improve treatment outcomes and quality of life in all affected patients. More caution and care will be required when applying dental-derived stem cells to the study and treatment of SCI due to spinal tumors in the future. This is because stem cell implantation risks tumor formation in the treatment of SCI [112]. In particular, stem cells, including iPSCs, may be tumorigenic, leading to the formation of teratomas and true tumors [113].

### 5.2. Future Research Direction

To understand dental stem cells’ potential in SCI treatment, future research should focus on key directions.

Large-animal studies and clinical trials are crucial for establishing robust clinical evidence. Comparative studies on different spinal cord injury models and their controllability should be considered [99]. Extensive comparative investigations into distinctions between dental stem cell sources and their efficacy across varying SCI severity levels will optimize stem cell selection. Exploring extracellular vesicles from various sources and optimizing therapy protocols is necessary [104]. Enhancements in the design and characteristics of scaffold materials combined with dental stem cell application should be pursued to improve biocompatibility, controlled and sustained growth factor release, and, ultimately, treatment effectiveness [92,100]. Research into secretomes produced by dental stem cells during SCI repair should be conducted to comprehend their role in tissue regeneration [94].

Furthermore, Biological mechanisms associated with dental stem cells need exploration to understand their role in neural regeneration [77,83,110] Maximizing the therapeutic potential of dental stem cells, effective transplantation methods, timing, and treatment regimens to enhance reparative outcomes, and optimization of differentiation through growth factors require attention [94]. This may require the optimization of differentiation through the use of growth factors and other differentiation-inducing factors [93]. As spinal cord injuries necessitate multiple modalities, research into the combined effects of various approaches, including rehabilitation therapy and pharmacological treatments, is pragmatic [85]. Improved control and adjustment of dental stem cell treatment outcomes for personalized treatment warrant further exploration [78]. Future studies may concentrate on alternative therapies combined with stem cell therapy, such as managing neurite growth inhibitors or using anti-Nogo A antibodies [105].

### 5.3. Exploration of Critical Questions

For extensive clinical applications, key issues must be addressed, including standardizing dental stem cell collection, expansion, and application procedures for consistent outcomes [106]. Research is needed to determine optimal delivery methods and timeframes for maximizing neural regeneration. An in-depth exploration of the interaction between dental stem cells and the immune system is essential to mitigate rejection risks and enhance treatment efficacy [80,98].

In summary, dental stem cells hold tremendous potential in SCI treatment, but further research is required to address existing limitations and ascertain the optimal strategies for clinical implementation. These efforts will contribute to enhancing the quality of life for SCI patients and realizing broader clinical applications.

## 6. Conclusions

Dental-derived stem cells represent a promising avenue for SCI therapy that can enhance the quality of life and clinical outcomes for SCI patients. Research has demonstrated their effectiveness in addressing SCI using different treatment strategies and plans. However, the application of dental-derived stem cells for SCI treatment faces technical challenges such as stem cell selection, maintenance of multipotency, cell fate determination, efficacy of delivery systems, and assessment of treatment outcomes. Further research is necessary to optimize treatment protocols and explore differences between various SCI models and biological mechanisms associated with dental-derived stem cells to maximize the therapeutic potential in future clinics.

## Figures and Tables

**Figure 1 biomedicines-12-00683-f001:**
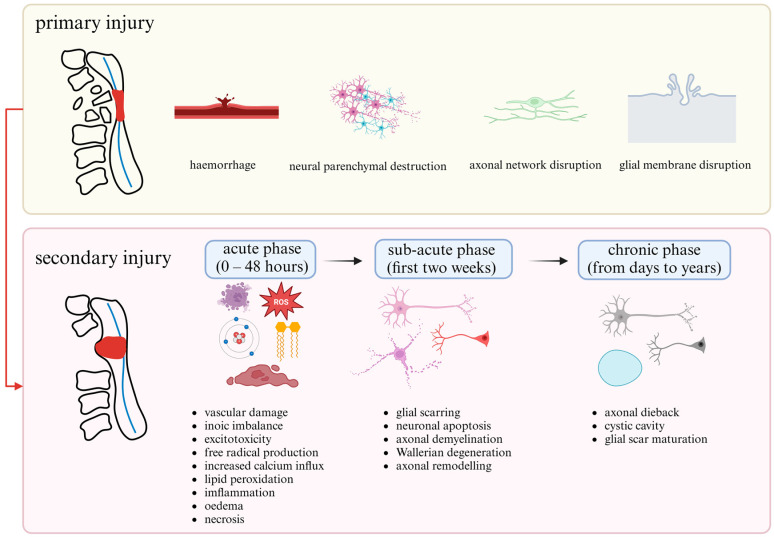
Phases and Pathophysiology of SCI. Created with https://www.biorender.com/ (accessed on 28 February 2024).

**Figure 2 biomedicines-12-00683-f002:**
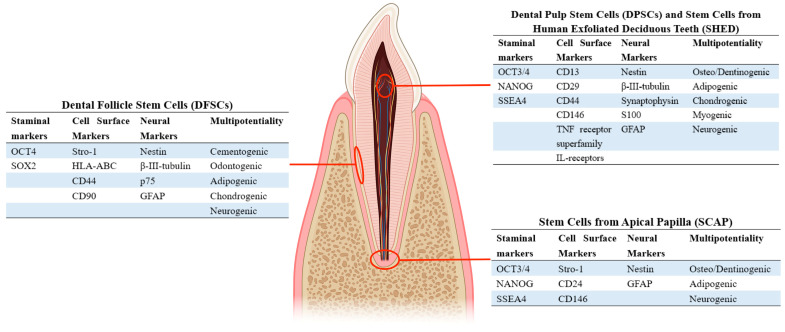
Anatomical Positions and Characteristics of Dental-Derived Stem Cells. Dental-derived stem cells, including DPSCs, DFSCs, SCAP, and SHED, exhibit diverse anatomical locations. Classification and neurogenic induction properties primarily rely on the expression of neuronal markers on cell surfaces, demonstrating multipotentiality. Created with https://www.biorender.com/ (accessed on 28 February 2024).

**Figure 3 biomedicines-12-00683-f003:**
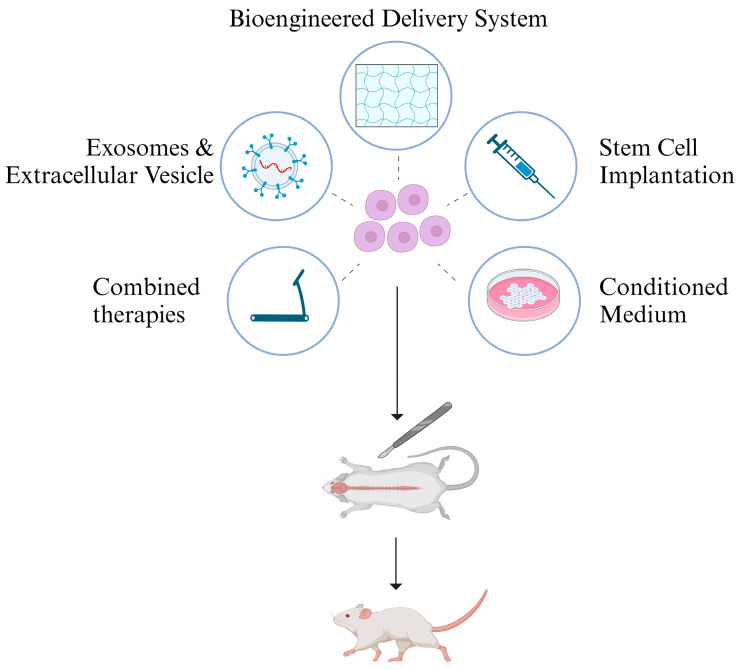
Multifaceted Applications of Oral-Derived Stem Cells in Spinal Cord Injury (SCI) Treatment. This illustration represents therapeutic strategies involving dental-derived stem cells, including stem cell transplantation, conditioned culture medium injection, innovative stem cell delivery systems, exosome-based therapies, and their combination with physical interventions. These strategies synergistically contribute to SCI recovery. Created with https://www.biorender.com/ (accessed on 28 February 2024).

**Table 1 biomedicines-12-00683-t001:** Characteristics of different SCI models.

Models	Characteristics	Refs.
Contusion	Extensive tissue pathology; White matter apoptosis; Demyelination Incomplete remyelination; Robust macrophage response Early cell apoptosis; Presence of cavities and fibrotic scars Changes extending several millimetres cranial and caudal to the injury epicentre	[25,26,27]
Compression	Widespread inflammation; Edema; Bleeding Ischemia and demyelination due to venous congestion Changes occur near the injury epicentre	[28,29]
Distraction	No significant vascular damage Membrane damage to neuronal cell bodies and axons extends several vertebral segments rostrally Extracellular space enlargement and white matter structural alterations	[30,31]
Dislocation	Intramedullary bleeding; Early cell apoptosis Membrane damage to neuronal cell bodies and axons extends several vertebral segments rostrally Extensive loss of nerve fibres and accumulation of β-amyloid precursor protein; Greatest loss in ventral and dorsal horn neuron	[27,30,31]
Transaction	Focal tissue pathology with white matter apoptosis; Demyelination Incomplete remyelination; Robust macrophage response at the injury epicentre Absence of apoptosis and demyelination at a distance from the epicentre Damage to the dural membrane, epidural hematoma, and leakage of cerebrospinal fluid due to knife wound, potentially leading to infection	[26,32]
Chemical	Ischemia; Demyelination; Oxidative damage (lipids and proteins) Inflammation; Cellular injury	[33]

**Table 2 biomedicines-12-00683-t002:** Advantages and disadvantages of dental stem cell application in SCI treatment.

Approach	Advantages	Disadvantages	The Role of SCI Repair	Refs.
Stem Cell Transplantation	Easy isolation, Minimal ethical controversy lower immunogenicity risk	Low stem cell survival Risk of immune rejection Cell dedifferentiation Risk of potential tumour formation	Promoting nerve regeneration. Improving motor function. Reducing spinal cord tissue loss. Protecting motor neurons. Mitigating inflammation. Decreasing neuronal apoptosis.	[77,78,79,80,81,82,108]
Conditioned Medium Injection	Contains growth factors for effective tissue repair. Avoid tumorigenesis and immune issues associated with stem cell transplantation.	The rapid diffusion of culture medium may be uncontrollable.	Establishing a favourable repair microenvironment. Promoting nerve regeneration.	[83,84,85]
Delivery System Approaches	Provide a suitable environment for stem cell survival, growth, and differentiation. Protects existing cells from apoptosis/necrosis. Allows controlled release of growth factors.	Material instability and potential quick degradation. Risk of cytotoxicity.	Enhancing stem cell viability and neural differentiation. Promoting neuronal regeneration. Inhibiting inflammation. Improving oxygen supply to damaged areas. Reducing gelatinous scarring to aid axonal and vascular regeneration.	[85,88,89,91,92,93,94,95,96,97,98,99,100,101]
Exosomes and Extracellular Vesicle	Exosomes exhibit most of the biological properties of stem cells. Exosomes are small and less likely to block microvessels. Exosomes have a low tumour risk.	Limited exosome production. Sensitivity to microenvironment pH.	Reducing inflammation and nerve damage. Improving spinal cord neuron survival. Enhancing motor function.	[104,109]
Combined Therapies	Electroacupuncture, treadmill training, and physiotherapy have shown some potential for improving functional recovery after SCI.	The timing and intensity of training can affect recovery outcomes.	No significant improvement in combination with treatment.	[105,106,107]

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
