# Peer review of "Advancements in Spinal Cord Injury Repair: Insights from Dental-Derived Stem Cells"

_biomedicines, 2024, doi:10.3390/biomedicines12030683_

Round 1

Reviewer 1 Report

Comments and Suggestions for Authors

This is a review article about potentials of dental-derived stem cells for treatment of spinal cord injury. To the most paper is comprehensive and well written. Illustrations are appropriate. My suggestions would be to:

1. Make a more elaborate description of the pathophisiology of spinal cord injury, particularly of various injury models used in various experimental studies.

2. The conclusions should be more structured, with clearly stated onfirmed results and aims for further expeirments.

Comments on the Quality of English Language

English is mostly fine, a spellcheck is necessary.

Reviewer 2 Report

Comments and Suggestions for Authors

In this comprehensive review, the authors present an overview of recent advancements in dental-derived mesenchymal stem cell therapy for spinal cord injury. The article is well-structured, incorporating pertinent figures and tables, rendering it informative for both researchers and physicians. Nevertheless, the following suggestions aim to enhance its overall value.

1.             The article currently addresses the underlying mechanisms separately, potentially causing confusion within the "treatment approach" section. Since understanding the underlying mechanism is crucial in evaluating the pros and cons of each method, especially for physicians, it is imperative to introduce a dedicated section summarizing these mechanisms. A visual aid, such as a figure illustrating the relationships and mechanisms of the treatment approaches, would significantly enhance the clarity of the content.

2.             Reference to the currently available treatments for spinal cord injury (SCI) in Line 58-59 requires scrutiny. As of my knowledge, no guidelines strongly endorse surgical intervention or pharmacological treatment as the primary method for treating spinal cord injury itself. While some research reports suggest their utility, the majority remain controversial. Therefore, it is recommended that the authors revise this section by providing appropriate references to accurately reflect the current status of these treatments.

Reviewer 3 Report

Comments and Suggestions for Authors

1.No reference on Scar tissue. Scar is present on the damaged level.

2. No ASIA scale proof before and after.

3. Totally different outcome between SCI and tumours.

4. No imaging proofs.

Round 2

Reviewer 2 Report

Comments and Suggestions for Authors

The authors appropriately revised and I can recommend for publication this article now.